# Minimal Residual Disease Detection at RNA and Leukemic Stem Cell (LSC) Levels: Comparison of RT-qPCR, d-PCR and CD26+ Stem Cell Measurements in Chronic Myeloid Leukemia (CML) Patients in Deep Molecular Response (DMR)

**DOI:** 10.3390/cancers15164112

**Published:** 2023-08-15

**Authors:** Elisabetta Abruzzese, Monica Bocchia, Malgorzata Monika Trawinska, Donatella Raspadori, Francesco Bondanini, Anna Sicuranza, Paola Pacelli, Federica Re, Alessia Cavalleri, Mirko Farina, Michele Malagola, Domenico Russo, Paolo De Fabritiis, Simona Bernardi

**Affiliations:** 1Hematology Unit, S. Eugenio Hospital, ASL Roma 2, Tor Vergata University, 00144 Rome, Italy; malgorzatamonika.trawinska@aslroma2.it (M.M.T.); paolo.defabritiis@aslroma2.it (P.D.F.); 2Chair of Hematology, University of Siena, Azienda Ospedaliera Universitaria, 53100 Siena, Italy; bocchia@unisi.it (M.B.); raspadori@unisi.it (D.R.); sicuranza4@unisi.it (A.S.); paola.pacelli@unisi.it (P.P.); 3Laboratory Medicine Unit, S. Eugenio Hospital, ASL Roma 2, 00144 Rome, Italy; francesco.bondanini@aslroma2.it; 4Bone Marrow Transplant Unit, ASST-Spedali Civili di Brescia, Chair of Hematology, Department of Clinical and Experimental Sciences, University of Brescia, 25123 Brescia, Italy; federica.re@unibs.it (F.R.); alessia.cavalleri@unibs.it (A.C.); m.farina004@unibs.it (M.F.); michele.malagola@unibs.it (M.M.); domenico.russo@unibs.it (D.R.)

**Keywords:** chronic myeloid leukemia, minimal residual disease, *BCR::ABL1*, RT-qPCR, digital PCR, leukemic stem cells, CD26

## Abstract

**Simple Summary:**

The monitoring of the minimal residual disease (MRD) in Chronic Myeloid Leukemia is a key element in the management of affected patients because it reflects the response to therapy and allows for the selection of the best responders who may benefit from the suspension of the therapy. The gold-standard method of MRD monitoring is the quantification of the *BCR::ABL1* transcript, the hallmark of the disease, by RT-qPCR. Considering that almost half of the patients who discontinue the treatment experience a molecular relapse, the identification of new approaches for the improvement of the selection of the best-responding CML patients is needed. In the present pilot study, we compared the gold standard with two additional MRD techniques: the quantification of the *BCR::ABL1* transcript by digital PCR and the quantification of leukemic stem cells by flowcytometry. In fact, this cell population is one of the factors driving relapses. Although no linear regression was observed, a correlation between the lowest levels obtained using the three methods was noted. To the best of our knowledge, this is the first time that these methods have been compared in the CML setting.

**Abstract:**

A Deep Molecular Response (DMR), defined as a *BCR::ABL1* transcript at levels ≤ 0.01% by RT-qPCR, is the prerequisite for the successful interruption of treatment among patients with Chronic Myeloid Leukemia (CML). However, approximately 50% of patients in Treatment-Free Remission (TFR) studies had to resume therapy after their *BCR::ABL1* transcript levels rose above the 0.1% threshold. To improve transcript detection sensitivity and accuracy, transcript levels can be analyzed using digital PCR (dPCR). dPCR increases *BCR::ABL1* transcript detection sensitivity 10–100 fold; however, its ability to better select successful TFR patients remains unclear. Beyond the role of the immune system, relapses may be due to the presence of residual leukemic stem cells (LSCs) that are transcriptionally silent. Flow cytometry can be used to identify and quantify circulating bone marrow Ph+ LSCs CD34+/CD38− co-expressing CD26 (dipeptidylpeptidase-IV). To date, the significance of circulating Ph+ LSCs in TFR is unclear. The aim of this work is to compare and examine the values obtained using the three different methods of detecting minimal residual disease (MRD) in CML at RNA (RT-qPCR and dPCR) and LSC (flowcytometry) levels among patients in TFR or exhibiting a DMR. The twenty-seven patients enrolled received treatment with either imatinib (12), dasatinib (6), nilotinib (7), bosutinib (1), or interferon (1). Twelve patients were in TFR, while the rest exhibited a DMR. The TFR patients had stopped therapy for less than 1 year (3), <3 years (2), 6 years (6), and 17 years (1). Blood samples were collected and tested using the three methods at the same time. Both d-PCR and LSCs showed higher sensitivity than RT-qPCR, exhibiting positive results in samples that were undetectable using RT-qPCR (17/27). None of the patients tested negative with d-PCR; however, 23/27 were under the threshold of 0.468 copies/μL, corresponding to a stable DMR. The results were divided into quartiles, and the lowest quartiles defined the lowest MRD. These data were strongly correlated in 15/27 patients, corresponding to almost half of the TFR patients. Indeed, the TFR patients, some lasting up to 17 years, corresponded to the lowest detectable DMR categories. To the best of our knowledge, this is the first attempt to analyze and compare DMRs in a CML population using standard (RT-qPCR) and highly sensitive (dPCR and LSCs) methods.

## 1. Introduction

A t(9; 22) (q34; q11) balanced reciprocal translocation is the cause of the chimeric BCR-ABL1 oncoprotein and generates chronic myeloid leukemia (CML), a clonal myeloproliferative disorder. The Philadelphia (Ph) chromosome and the *BCR-ABL1* oncogene are generated by this rearrangement. The disease marker is its chimeric transcript [1]. With a survival rate comparable to that of age-matched healthy people, CML patients can be successfully treated with tyrosine kinase inhibitor (TKI) therapy that targets BCR-ABL1-positive cells and causes hematologic and molecular remission in 80–90% of treated patients [2,3,4]. 

Since the successful introduction of Tyrosine Kinase Inhibitors (TKIs), the monitoring of the minimal residual disease (MRD) in Chronic Myeloid Leukemia (CML) adult cases has become pivotally important for patients’ management and therapeutic strategy choices.

MRD is conventionally monitored by quantifying *BCR::ABL1* transcript levels using RT-qPCR, normalized for the quantification of a housekeeper gene (e.g., *ABL1*), in peripheral blood (PB) cells. A stable Deep Molecular Response (DMR), defined as *BCR::ABL1* transcript levels ≤0.01% determined via RT-qPCR, is the prerequisite for a successful interruption of treatment among patients with CML. However, approximately 50% of patients enrolled in Treatment-Free Remission (TFR) studies had to resume therapy after their *BCR::ABL1* transcript levels rose above the 0.1% threshold, which is known as Major Molecular Remission (MMR). These events are experienced regardless of other variables like the risk upon diagnosis, the duration of TKI therapy, the duration of the DMR, and so on. Thus, it appears that DMR quantified via RT-qPCR is not an optimal parameter for the selection of patients who can safely discontinue treatment and maintain TFR.

Digital PCR (dPCR) has recently emerged as a more precise and accurate technique for detecting MRD among CML patients by absolutely quantifying *BCR::ABL1* transcript levels. This technique is based on the partitioning of a reaction mix into thousands of micro-PCR reactions. Amplification, independently of the dPCR platform (which is either chip-based or droplet-based), reveals the presence of the target through fluorescence, and this allows for the absolute quantification through the counting of the positive micro-reactions. Moreover, it seems to surpass the sensitivity of RT-qPCR by 10–100 fold and its accuracy as well [5], thus increasing the interest in its use in clinical practice [6]. At present, different studies are investigating whether dPCR may also help in the better identification of patients who will not relapse after discontinuation of TKI therapy by taking advantage of TFR [7,8,9,10,11].

However, the possibility that a relapse after the discontinuation of TKI administration may be due to the persistence of leukemic stem cells (LSCs) that are insensitive to TKIs renders even the most sensitive PCR-based assay for monitoring *BCR::ABL1* transcript levels unable to monitor the residual quiescent CML LSCs. In fact, LSCs may present very low levels of transcripts or even be transcriptionally silent while also surviving indefinitely in tumor-specific hypoxic niches and circulating in the PB. Hematopoietic stem cells are characterized by the CD34+/CD38−/Lin− immunophenotype. CD26 (dipeptidylpeptidase IV) has been identified as a biomarker for the quantification and isolation of CML LSCs in the bone marrow (BM) of CML patients, as CD26+-sorted LSCs have been demonstrated to be entirely *BCR-ABL1*-positive at the genomic level. Recently, LSCs have been successfully quantified in PB samples as well, and the use of PB CD26+ LSCs flow cytometry assays appears to be feasible, reproducible, specific, and sensitive [12,13]. This makes the latter method suitable for routine monitoring.

In the present preliminary study, we compared and examined the results obtained using three different methods of detecting MRD in CML: (i) *BCR::ABL1* transcript quantification by RT-qPCR, (ii) the quantification of the same transcriptby dPCR, and (iii) the quantification of circulating LSC in patients in TFR or exhibiting a DMR.

## 2. Materials and Methods

Twenty-seven adult patients afflicted with CML were enrolled in two biological, clinical, non-pharmacological trials, namely, “AIRC IG 20133” and “NP-DPCR-CML_1603”, approved by the Ethics Committees of Siena and Brescia (Italy), respectively. The studies were conducted in accordance with the Declaration of Helsinki, and the enrolled patients provided their written informed consent. They received treatment with either imatinib (12; 44%), dasatinib (6; 22%), nilotinib (7; 26%), bosutinib (1; 4%), or interferon (1; 4%). Upon enrollment, 12 patients (44%) were in TFR, while the rest (66%) exhibited a DMR. The TFR patients had stopped receiving therapy for less than 1 year (3; 25%), <3 years (2; 17%), 6 years (6; 50%), and 17 years (1; 8%). A total of 10 mL of Peripheral Blood (PB) was collected using EDTA tubes and centrifuged in order to obtain the buffy coat within 24 h of sampling for RT-qPCR analysis. A total of 5 mL of fresh PB from the samples was sent to Azienda Ospedaliera Universitaria in Siena for LSC CD26+ quantification. A total of 10 mL of fresh PB collected in EDTA tubes was sent to ASST Spedali Civili of Brescia for dPCR analysis. All the shipments were carried out by an express courier within 24 h.

RNA extraction was performed after the PB samples were centrifuged to obtain PB cells using NucleoSpin RNA plus (Machery Nagel, Duren, Germany), as specified in the manufacturers’ instructions. This procedure has been approved for clinical diagnostic application in Italy. Extracted RNA was immediately quantified using a Qubit RNA high-sensitivity kit (ThermoFisher Scientific, Waltham, MA, USA) and retrotranscribed using the following method.

A total of 1 μg of total RNA in 10 μL of DNA- and RNA-free H_2_O was retrotranscribed after being incubated at 70 °C for 10′, followed by ice incubation. Then, 100 U of Reverse transcriptase (Superscript I or II), 1 mM of dNTP, 10 mM of DTT, 25 μM of Random hexamers, and 20 U of RNAse inhibitor and RT buffer (according to the enzyme used) were added to a final volume of 20 μL. This step was followed by three subsequent incubations: at room temperature for 10′, at 42 °C for 45′, and at 99 °C for 3′, with a final holding period at 4 °C. The cDNA was quantified using Qubit ssDNA kit (ThermoFisher Scientific) and then stored at −20 °C until analysis. Retrotranscription was performed as described in Bernardi S. et al. (2019) [7] in order to guarantee the normalization of the amount of RNA considered during the analysis.

*BCR::ABL1* transcript quantification was performed using the automated Xpert Ultra BCR-ABL1 MonitorTM Cepheid method and calibrated for the transcript of the *ABL1* reference gene and the *BCR-ABL1* target gene, with a PCR sensitivity of 5.0 (>250,000 *ABL1* copy numbers). Samples were analyzed according to the manufacturer’s instructions and by following the internationally standardized methods of MRD monitoring for CML. Accordingly, molecular responses were expressed and reported as *BCR::ABL1* % on a log scale, where 0.1%, 0.01%, 0.0032%, and 0.001% corresponding to MR3.0, MR4.0, MR4.5, and MR5.0, respectively. The minimum sum of the *ABL1* reference gene transcripts, irrespective of whether *BCR::ABL1* was detected, should be 10.000 for MR4.0, 32.000 for MR4.5, and 100.000 for MR5.0. In fact, DMR classes may be assessed even in the case of undetectable *BCR::ABL1* transcript levels (U). In this case, the MR classes are determined according to the *ABL1* transcript copy number, i.e., 10.000, 32.000, and 100.000 for MR4.0, MR4.5, and MR5.0, respectively [14].

d-PCR analysis was performed using a TaqMan-MGB probe targeting the *BCR::ABL1* transcript. The entire procedure was conducted as previously published [7,15], and the sensitivity threshold and controls were established as presented and validated in the research by Fava C. et al. (2021) [16]. A custom assay was designed using a FAM-label based on the sequence of routinely used probes. *BCR::ABL1* quantifications were performed by analyzing 50 ng of cDNA using a QuantStudio 3D Digital PCR System (ThermoFisher Scientific). The fixed quantity of cDNA considered during the analysis allowed for the normalization of the results and the non-saturation of the instrument, as previously reported by our group [7]. *BCR::ABL1* transcript values determined using dPCR were expressed as numbers of copies/μL. The secondary analysis was carried out using AnalysisSuite Cloud Software (V 3.1) (ThermoFisher Scientific). A value of 0.468 copies/μL of *BCR::ABL1* transcript is a previously calculated cut-off capable of identifying patients presenting optimal response to TKIs [7]. In particular, patients presenting *BCR::ABL1* transcript levels above the cut-off present a significantly lower probability of obtaining stable TFR than patients presenting *BCR::ABL1* transcript levels below the cut-off.

To detect circulating CD34+/CD38−/CD26+ LSCs, PB cells of CML patients were incubated with a mixture of pre-titred lyophilized antibodies containing anti-CD45 (BD Biosciences, Heidelberg, Germany), anti-CD34 (581), anti-CD38 (HIT2), and anti-CD26 (M-A261) (BD Bioscience). Isotype controls were included in each staining procedure. Acquisition and analysis were performed using a FACSCanto II flow cytometer coupled with FACSDiva software (Ver 8.0) (BD Biosciences, NR Nannini). To obtain a sensitivity comparable with and superimposable to molecular assays, 1,000,000 CD45+ events were acquired. Median absolute number of CD26+ cells/μL was calculated as follows: (# WBCs/μL) × (% CD34+/CD38−/CD26+ stained CD45+ cells) [12,13].

A descriptive statistical analysis was conducted in order to evaluate the demographic and clinical variables, such as the therapy and the TFR. A linear regression analysis was performed to evaluate the correlation between the results obtained via RT-qPCR, dPCR, and CD26+ cell quantification. Moreover, the data were subdivided into quartiles for their analysis. Statistical analysis was performed using GraphPad (Ver. 9.5.1).

## 3. Results

The TFR status, the type of TKI treatment, the RT-qPCR results, the dPCR data, and LSCs quantification were employed as variables, and their details are summarized in Table 1. 

Considering the entirety of the results, the RT-qPCR values ranged from 0–0.0068 (%IS), the d-PCR values ranged from 0.073 to 0.943 (*BCR::ABL1* copies/μL), and the LSC values ranged from 0 to 0.156 (cells/μL). Both d-PCR and LSCs showed higher sensitivity than RT-qPCR, exhibiting positive results in samples presenting undetectable levels of *BCR::ABL1* transcript by RT-qPCR (17/27, 63%). None of the patients tested negative via d-PCR. However, 23/27 (85%) were under the threshold of 0.468 copies/μL of *BCR::ABL1* transcript, which has previously been described as corresponding to a “true” DMR. LSC levels were negative in six patients (22%), four of which (4/6, 66%) also tested negative by RT-qPCR. 

No linear regression was found by comparing RT-qPCR vs. dPCR, dPCR vs. CD26+ cells, nor RT-qPCR vs. CD26+ cells.

The results obtained using all the analyses were divided into five levels corresponding to a range from undetectable measurements up to one target for the first level and four quartiles for the other determination (Table 2). The subdivisions were based on molecular responses regarding RT-qPCR (undetectable, MR5.0, MR4.5, MR4.0, and MR3.0), the copies/μL absolute quantification with respect to dPCR (where <0.99 denotes one copy; <0.240 denotes two to three copies; <0.280 denotes four copies; <0.468 denotes five copies, and >0.468 denotes more than six copies), and the distribution for the LSCs (Negative, denoting 0 cells; >0.005 <0.04; >0.04 <0.07; >0.07 <1; >1 cells/μL). 

Considering the subdivision into five levels, only 3/27 (11%) samples presented total concordance among the three methods (Table 3A). In addition, 5/27 (18.5%), 3/27 (11%), and 4/27 (14.8%) presented concordance only between RT-qPCR and dPCR, dPCR and CD26+ cell number, and RT-qPCR and CD26+ cell number, respectively (Table 3B, C, and D). The highest concordance was observed between methods quantifying the same molecular target: the *BCR::ABL1* transcript (Table 3B).

The two lowest quartiles, together with the levels corresponding to undetectability, defined the lowest detectable DMR. Considering these levels altogether, a strong correlation of these data was noted in 15/27 patients. Among them, 7/15 (46.6%) concordant cases were in TFR, corresponding to 58.3% of the total patients in TFR enrolled in this study. Among the 26 DMR patients, defined using the gold standard, 11 (42%) were discordant. 

## 4. Discussion

The improvement of MRD monitoring for adult CML patients is one of the main goals of the current CML management strategy. Until a few years ago, the quantitative reverse polymerase chain reaction (RT-qPCR) determination of *BCR::ABL1* transcript levels was the gold-standard method for monitoring minimal residual disease (MRD) in CML and for the best management of CML patients [17]; nowadays, the limits of RT-qPCR in quantifying *BCR::ABL1* transcript levels have been widely reported along with their overcomes via dPCR. In fact, in recent years, the molecular surveillance of MRD in hematological malignances has been transformed by dPCR. In addition, the inability to detect LSCs that are still present in the BMs of patients who have sustained undetectable molecular residual disease is one of the major limitations of CML MRD surveillance [18]. Particular types of stem cells, for example, some pre-leukemic neoplastic stem cells, may be resistant to TKI therapy because they are slow-cycling cells with various stem cell resistances, but the majority of LSCs and their sub-clones may be receptive to TKI therapy [19,20,21,22]. Since CD26 is not observed in normal SCs or in LSCs from other hematological malignancies, it may be thought as a robust, specific marker of CML LSCs, but little is known about circulating CD26+ LSCs; their role, if they even play one, as an active reservoir of disease; or their potential significance in terms of MRD [23]. To the best of our knowledge, this study is the first attempt to analyze and compare DMR in a monocentric CML population using standard (RT-qPCR) and highly sensitive methods, such as *BCR::ABL1* transcript quantification via dPCR and LSC CD26+ detection and quantification. The enrolled patients had been undergoing different treatments. In particular, 15/27 patients were receiving TKI therapy, while 12/27 were in TFR. Among the TFR patients, some lasting up to 17 years, almost half were in the lowest-detectable DMR categories by all three MRD techniques. This is a notable result since the enrolled TFR patients presented at least 2 years of sustained TFR. The reported timing of molecular relapse after the discontinuation of TKI administration was within 1 year from therapy suspension [24]. Indeed, the concordance of the three techniques in detecting low levels of MRD may be related to a real MRD control, even if we were not able to observe any linear correlation considering the entire cohort of patients. Another important observation is that the major rate of concordance was observed between techniques detecting the same target, namely, the *BCR::ABL1* transcript. The RT-qPCR and dPCR approaches reported in the present manuscript consider a relative and an absolute quantification of *BCR::ABL1* transcript, respectively. This is a very important point since *BCR::ABL1* is conventionally quantified by considering normalization with a reference gene in order to confirm the quality of the pre-analytic steps and identify the MR classes following the IS. In the case of the presented dPCR quantification, *BCR::ABL1* was absolutely quantified, and the comparability of the results was facilitated by the normalization of the quantity of RNA retrotranscribed and by the quantity of cDNA analyzed via dPCR, as stated in the Materials and Methods section. Moreover, dPCR is less sensitive than RT-qPCR to nucleic acid degradation thanks to its ability to amplify short sequences detected by MGB probes. Notably, the aim of *BCR::ABL1* dPCR quantification is the improvement of accuracy and sensitivity, not the simple transfer of the IS from RT-qPCR to a new technology. Despite this difference, the two molecular techniques turned out to be the most concordant in the identification of patients presenting very low MRD levels. In part, this is not surprising since leukemic stem cells are known to be almost quiescent and non-expressing *BCR-ABL1*. In this context, a recent study aiming to evaluate CD26 expression in suspected cases of CML in the chronic phase and correlating CD26+ stem cells at baseline and their kinetics with tyrosine kinase inhibitor (TKI) treatment on a further follow-up at 12 months led to an interesting result [25]. Of the seven patients included in the study, specifically the only patients with available follow-up data and who had achieved major molecular remission with TKIs, two tested negative for CD26+ LSCs, while five showed a significant drop in the level of CD26+ LSCs, which was, however, still detectable via flow cytometry. Thus, the decline in LSC CD26+ levels during TKI administration merits further research. Very little is known about the biological implications of circulating LSCs in CML. According to standard definitions, leukemic stem cells (LSC) are a population of cells that creates and maintains a leukemic clone [26]. LSCs in CML are classically considered as the result of an HSC acquiring *BCR-ABL1* [19,27]. The BCR-ABL1 protein cannot, however, cause leukemia on its own [20]. Certainly, larger studies and dynamic scoring will help define their informative and predictive value. Moreover, considering the suggestive observations reported in the present report, further investigation of the correlation between circulating CD26+ LSCs and the *BCR::ABL1* transcript levels relating to the identification of new CML patient sub-settings is warranted.

## Figures and Tables

**Table 1 cancers-15-04112-t001:** MRD results per patient. TFR = treatment-free remission; U = undetectable.

Treatment	TFR	RT-qPCR IS%	dPCR Copies/μL	CD26+ LSC/μL
Nilotinib	NO	0 (U)	0.497	0
Interferon	TFR > 10y	0 (U)	0.0764	0
Nilotinib	NO	0.00098	0.943	0.0102
Dasatinib	NO	0 (U)	0.384	0.0054
Bosutinib	NO	0.00047	0.164	0.0307
Imatinib	TFR > 5y	0 (U)	0.156	0.0413
Nilotinb	NO	0 (U)	0.088	0.0088
Imatinib	NO	0.0068	0.0757	0.1194
Imatinib	TFR > 5y	0 (U)	0.0823	0
Imatinib	NO	0.0013	0.078	0
Imatinib	TFR > 5y	0 (U)	0.162	0.0385
Nilotinb	NO	0.0044	0.251	0.026
Nilotinb	TFR > 2y	0.0019	0.477	0.0459
Dasatinib	NO	0 (U)	0.0842	0.0845
Nilotinb	NO	0 (U)	0.833	0.052
Dasatinib	NO	0 (U)	0.220	0.0694
Dasatinib	TFR > 3y	0 (U)	0.078	0.0505
Imatinib	TFR > 3y	0.0033	0.0731	0.0412
Dasatinib	NO	0.00055	0.146	0
Imatinib	TFR > 2y	0 (U)	0.166	0.0506
Imatinib	TFR > 5y	0.029	0.402	0.0271
Dasatinib	NO	0 (U)	0.258	0.0215
Imatinib	NO	0 (U)	0.0852	0.0544
Imatinib	TFR > 5y	0 (U)	0.164	0.1565
Nilotinib	TFR > 2y	0 (U)	0.423	0.0513
Imatinib	NO	0.00066	0.0742	0.0684
Imatinib	TFR > 5y	0 (U)	0.158	0

**Table 2 cancers-15-04112-t002:** Raw data regarding the quantification of *BCR::ABL1* transcript via RT-qPCR (A), dPCR (B), and CD26+ LSC via flow-cytometry (C). Results are ordered from lowest to highest. The different colors signify the different quartiles, with the darker colors representing the higher quartiles. Green = undetectable via RT-qPCR, <0.99 denotes 1 copy detected via dPCR, and negative denotes 0 cells detected via flowcytometric LSC quantification; yellow = MR5.0 via RT-qPCR, >0.99 <0.240 denotes 2–3 copies detected via dPCR, and >0.005 <0.04 cells/μL detected via flowcytometric LSC quantification; orange = MR4.5 via RT-qPCR, >0.240 <0.280 denotes 4 copies detected via dPCR, >0.04 <0.07 cells/μL via Flowcytometric LSC quantification; red = MR4.0 via RT-qPCR, >0.280 <0.468 denotes 5 copies detected via dPCR, and >0.07 <1 cells/μL via Flowcytometric LSC quantification; grey = MR3.0 via RT-qPCR, >0.468 denotes more than 6 copies detected via dPCR, and >1 cells/μL via Flowcytometric LSC quantification.

A		B		C
**RT-qPCR IS%**		**dPCR Copies/μL**		**CD26+ LSC/μL**
0		0.0731		0
0		0.0757		0
0		0.0764		0
0		0.078		0
0		0.078		0
0		0.0823		0.0054
0		0.0842		0.0088
0		0.0852		0.0102
0		0.088		0.0215
0		0.146		0.026
0		0.156		0.0271
0		0.158		0.0307
0		0.162		0.0385
0		0,164		0.0412
0		0.164		0.0413
0		0.166		0.0459
0.00047		0.220		0.0505
0.00055		0.251		0.0506
0.00066		0.258		0.0513
0.00098		0.384		0.052
0.0013		0.402		0.0544
0.0019		0.423		0.0684
0.0033		0.477		0.0694
0.0044		0.497		0.0845
0.0068		0.833		0.1194
0.029		0.943		0.1565

**Table 3 cancers-15-04112-t003:** Samples presenting concordance between different methods. (**A**) Reported data regarding samples presenting concordance between all the considered methods. (**B**) Reported data regarding samples presenting concordance between RT-qPCR and dPCR. (**C**) Reported data regarding samples presenting concordance between dPCR and CD26+ cell number. (**D**) Reported data regarding samples presenting concordance between RT-qPCR and CD26+ cell number. TFR = treatment-free remission; LSC = leukemic stem cells. Green = undetectable via RT-qPCR, <0.99 denotes 1 copy detected via dPCR, and negative denotes 0 cells detected via flowcytometric LSC quantification; yellow = MR5.0 via RT-qPCR, >0.99 <0.240 denotes 2–3 copies detected via dPCR, and >0.005 <0.04 cells/μL detected via flowcytometric LSC quantification; orange = MR4.5 via RT-qPCR, >0.240 <0.280 denotes 4 copies detected via dPCR, >0.04 <0.07 cells/μL via Flowcytometric LSC quantification; red = MR4.0 via RT-qPCR, >0.280 <0.468 denotes 5 copies detected via dPCR, and >0.07 <1 cells/μL via Flowcytometric LSC quantification; grey = MR3.0 via RT-qPCR, >0.468 denotes more than 6 copies detected via dPCR, and >1 cells/μL via Flowcytometric LSC quantification.

**A**				
**Treatment**	**TFR**	**RT-qPCR IS%**	**dPCR Copies/μL**	**CD26+ LSC/μL**
Interferon	TFR > 10y	0 (U)	0.0764	0
Bosutinib	NO	0.00047	0.164	0.0307
Imatinib	TFR > 5y	0 (U)	0.0823	0
**B**				
**Treatment**	**TFR**	**RT-qPCR IS%**	**dPCR Copies/μL**	**CD26+ LSC/μL**
Nilotinib	NO	0 (U)	0.088	0.0088
Dasatinib	NO	0 (U)	0.0842	0.0845
Dasatinib	TFR > 3y	0 (U)	0.078	0.0505
Dasatinib	NO	0.00055	0.146	0
Imatinib	NO	0 (U)	0.0852	0.0544
**C**				
**Treatment**	**TFR**	**RT-qPCR IS%**	**dPCR Copies/μL**	**CD26+ LSC/μL**
Imatinib	NO	0.0013	0.078	0
Imatinib	TFR > 5y	0 (U)	0.162	0.0385
Imatinib	TFR > 3y	0.0033	0.0731	0.0412
**D**				
**Treatment**	**TFR**	**RT-qPCR IS%**	**dPCR Copies/μL**	**CD26+ LSC/μL**
Nilotinib	NO	0 (U)	0.497	0
Nilotinib	NO	0.00098	0.943	0.0102
Nilotinib	TFR > 2y	0.0019	0.477	0.0459
Imatinib	TFR > 5y	0 (U)	0.158	0

## Data Availability

Data are available upon request by the corresponding authors.

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
