# Peer review of "Minimal Residual Disease Detection at RNA and Leukemic Stem Cell (LSC) Levels: Comparison of RT-qPCR, d-PCR and CD26+ Stem Cell Measurements in Chronic Myeloid Leukemia (CML) Patients in Deep Molecular Response (DMR)"

_cancers, 2023, doi:10.3390/cancers15164112_

Round 1
Reviewer 1 Report
In this manuscript, Abruzzese et al. observe a relationship among three methods, RT-qPCR, d-PCR, and CD26+ stem cell flow cytometry. They found no linear regression neither comparing RT-qPCR vs dPCR, dPCR vs CD26+ cells, nor RT-qPCR vs CD26+ cells. This reviewer has the following concerns.
Major comments:
1. In Table 2, data from different samples are next to each other. If the authors hope to show the distribution of data from each method, data from RT-qPCR, dPCR, and CD26+ should be shown individually.
2. A new table that showed concordance between the 3 methods should be added, because it is hard to recognize that 5/27, 3/27, and 4/27 presented concordance only between RT-qPCR vs dPCR, dPCR vs CD26+ cells, and RT-qPCR vs CD26+ cells in Table 1.
Author Response
REVIEWER 1
In this manuscript, Abruzzese et al. observe a relationship among three methods, RT-qPCR, d-PCR, and CD26+ stem cell flow cytometry. They found no linear regression neither comparing RT-qPCR vs dPCR, dPCR vs CD26+ cells, nor RT-qPCR vs CD26+ cells. This reviewer has the following concerns.
Major comments:
- In Table 2, data from different samples are next to each other. If the authors hope to show the distribution of data from each method, data from RT-qPCR, dPCR, and CD26+ should be shown individually.
A1. We really thank the reviewer for the suggestion. We have elaborated Table 2 dividing the results obtained by the three methods. The present form of Table 2 is separated into Table 2A for RT-qPCR results, Table 2B for dPCR results, and Table 2C for LSC CD26+ quantification results. We hope that the new version avoids any confusion concerning the comparison of the data. Please, see Pages 6 and 7.
- A new table that showed concordance between the 3 methods should be added, because it is hard to recognize that 5/27, 3/27, and 4/27 presented concordance only between RT-qPCR vs dPCR, dPCR vs CD26+ cells, and RT-qPCR vs CD26+ cells in Table 1.
A2. We totally agree with the Reviewer. We have added a new table, namely Table 3, reporting the samples presenting concordance between the different methods. The new table is divided into 4 different sub-tables depending on the concordance. Please, see Table 3 at pages 8 and 9.
Reviewer 2 Report
- The recommendation for TFR in CML patients who achieve a deep molecular response is not older than 13-14 years. Could the authors please highlight how one of the patients in TFR has been off treatment for 17 years?
- There is inconsistency in the nomenclature for BCR::ABL1 in the text. Please ensure it is consistent.
- The RNA extraction and cDNA synthesis method were not highlighted in the method section. Was RNA measured for the synthesis of cDNA? Was the same cDNA source used for RT-qPCR and dPCR?
- Was any normalization done for the dPCR? Normalization has been well-established for RT-qPCR, which makes comparison among different samples possible. How was the normalization of the data for dPCR done? Did the author use the same cDNA source and the same volume of cDNA for dPCR and RT-qPCR?
- How was the normalization for the detection of LSC? The authors mentioned that, in order to obtain comparable sensitivity, they used 1,000,000 cells to study LSC, but no explanation is given on how this number of cells was determined and how this can be correlated to RT-qPCR or dPCR? Does 1000000 cells correlate to a particular number of ABL1 copies?
- Did the author use any normal control? Detection of BCR::ABL1 in samples from normal individuals has been reported when highly sensitive techniques are applied. Have the authors checked whether the detection frequency of positive BCR::ABL1 by dPCR in BCR::ABL1 negative samples by RT-qPCR is different from samples from normal individuals?
Author Response
REVIEWER 2
- The recommendation for TFR in CML patients who achieve a deep molecular response is not older than 13-14 years. Could the authors please highlight how one of the patients in TFR has been off treatment for 17 years?
A1. Thank you so much for this important question. The patient was diagnosed in 1994 and treated with IFN+AC, he rapidly reached a complete molecular response (1997) and continued on IFN. The clinicians decided to stop therapy after he suffered from heart problems in 2001, ready to restart therapy with TKI in case of increased MRD level. The patient has maintained CMR and is now considered in TFR.
- There is inconsistency in the nomenclature for BCR::ABL1 in the text. Please ensure it is consistent.
A2. We thank the Reviewer for this point. The text has been corrected and edited accordingly with the conventional nomenclature, meaning BCR-ABL1 referring to the protein, BCR-ABL1 in italic referring to the gene, and BCR::ABL1 referring to the transcript. Please, see all the corrections in the manuscript’s text.
- The RNA extraction and cDNA synthesis method were not highlighted in the method section. Was RNA measured for the synthesis of cDNA? Was the same cDNA source used for RT-qPCR and dPCR?
A3. Thank you for these questions. We added the information concerning the RNA extraction and cDNA synthesis in Materials and methods section. Both RNA and cDNA were quantified before analysis and the quantity of cDNA was used as normalisation factor for dPCR analysis, as previously published (please, see Bernardi at al, Cancer Medicine, 2019). The RT-qPCR analysis were performed by the enrolling centre, while LCS CD26+ quantification and dPCR analysis were performed by Siena and Brescia Groups, respectively. The management of the samples and their centralization has been better explained in the Materials and Methods section. Please, see page 3 from line 115 to line 132.
- Was any normalization done for the dPCR? Normalization has been well-established for RT-qPCR, which makes comparison among different samples possible. How was the normalization of the data for dPCR done? Did the author use the same cDNA source and the same volume of cDNA for dPCR and RT-qPCR?
A4. Thank you for these questions. The dPCR analysis was performed as previously published (Bernardi s et al, Cancers medicine 2019) and as set in the alignment study performed for inter-laboratories standardisation of dPCR platforms applied for BCR::ABL1 transcript quantification. Different quantities of cDNA were used for RT-qPCR (performed following the international guide lines for IS application) and dPCR (performed as reported in the references number 5, 7, 15, and 16).
- How was the normalization for the detection of LSC? The authors mentioned that, in order to obtain comparable sensitivity, they used 1,000,000 cells to study LSC, but no explanation is given on how this number of cells was determined and how this can be correlated to RT-qPCR or dPCR? Does 1000000 cells correlate to a particular number of ABL1 copies?
A5. Thank you for this question that enables us to better explain our approach. In order to obtain a sensitivity that is comparable to molecular biology assays, flow-cytometry analysis is performed acquiring at least 1,000,000 CD45+ leukocytes and applying a strict gating strategy that permit to detect the presence of CML CD26+ LSCs. It is essential to perform the test on proper samples, collecting at least 7mL peripheral blood on which a cell blood count is performed, followed by a centrifugation to collect the buffy coat. No correlations can be found between number of acquired CD45+ cells and ABL1 copies.
- Did the author use any normal control? Detection of BCR::ABL1 in samples from normal individuals has been reported when highly sensitive techniques are applied. Have the authors checked whether the detection frequency of positive BCR::ABL1 by dPCR in BCR::ABL1 negative samples by RT-qPCR is different from samples from normal individuals?
A6. Thank you so much for this important point. The dPCR sentitivity has been previously described as 1 copy of BCR-ABL1 per reaction vs 3 copies of BCR-ABL1 quantifiable by RT-qPCR in the reference 15. The identification of the sensitivity of the dPCR was assessed following the method “Standard Deviation of Blanks Response” as reviewed by Shrivastava A, Gupta VB. “Methods for the determination of limit of detection and limit of quantitation of the analytical methods.” Chron Young Sci 2011;2:21-5. This method is used for LoD determination when blanks standard deviation is a nonzero value and does not include the analysis of diluted positive samples. During the setup of the technique, different healthy controls were analyzed, while every dPCR run included a blank control.
Round 2
Reviewer 1 Report
All the comments have been addressed.
Author Response
Thank you so much for your previous suggestions that helped us in improving the manuscript.
Reviewer 2 Report
The authors have addressed the issues raised in the first round of review. However, there is one aspect that requires further explanation, namely the normalization of the BCR::ABL1 result. The authors cited their earlier paper from 2019, where they established the threshold for DMR using their ddPCR technique. In that paper, RNA was quantified, and a concentration of 50 ng/μl was used for cDNA synthesis. In this current paper, the authors mention that the cDNA synthesis was performed in the centers from where the samples originated. It is essential to clarify whether the same concentration rule was followed for cDNA synthesis in this study. If it was, this information should be explicitly mentioned in the text. On the other hand, if a different approach was taken, the authors should explain how they corrected for any discrepancies.
Another question that arises is why normalization using ABL1 quantitation was not performed. The authors might consider discussing this matter in the conclusion section. Originally, normalization to ABL1 was established to correct for differences in sample volume, RNA concentration, and other factors. Understanding the technical difficulties that led the authors to measure RNA and use the same amount for standardization, despite the potential variability in quantitation depending on the method of measurement, would be beneficial for readers, especially those involved in analyzing and reporting this test. Including a detailed discussion of these issues in the conclusion section would be highly valuable.
Author Response
We really thank the Reviewer for the comments that have helped us in improving our manuscript.
Please, find below the answers to the questions of the 2nd round of revision.
Q1. The authors have addressed the issues raised in the first round of review. However, there is one aspect that requires further explanation, namely the normalization of the BCR::ABL1 result. The authors cited their earlier paper from 2019, where they established the threshold for DMR using their ddPCR technique. In that paper, RNA was quantified, and a concentration of 50 ng/μl was used for cDNA synthesis. In this current paper, the authors mention that the cDNA synthesis was performed in the centers from where the samples originated. It is essential to clarify whether the same concentration rule was followed for cDNA synthesis in this study. If it was, this information should be explicitly mentioned in the text. On the other hand, if a different approach was taken, the authors should explain how they corrected for any discrepancies.
A1. Thank you so much for this point. We performed the dPCR analysis following our set up method, also in terms of pre-analytic steps. We add this information in the text of the manuscript. Please, see lines 135-138, and 157-160.
Q2. Another question that arises is why normalization using ABL1 quantitation was not performed. The authors might consider discussing this matter in the conclusion section. Originally, normalization to ABL1 was established to correct for differences in sample volume, RNA concentration, and other factors. Understanding the technical difficulties that led the authors to measure RNA and use the same amount for standardization, despite the potential variability in quantitation depending on the method of measurement, would be beneficial for readers, especially those involved in analyzing and reporting this test. Including a detailed discussion of these issues in the conclusion section would be highly valuable.
A2. Thank you so much for this point. We agree with the reviewer that the difference between a normalized and an absolute quantification must be stressed and underlined, in particular in the context of CML, where the MRD monitoring is historically based on a normalization to a reference gene. We stressed this aspect from line 265 to line 278.